# FusionFormer: A Multi-sensory Fusion in Bird's-Eye-View and Temporal Consistent Transformer for 3D Object Detection

## Abstract

Multi-sensor modal fusion has demonstrated strong advantages in 3D object detection tasks. However, existing methods that fuse multi-modal features require transforming features into the bird's eye view space and may lose certain information on Z-axis, thus leading to inferior performance. To this end, we propose a novel end-to-end multi-modal fusion transformer-based framework, dubbed FusionFormer, that incorporates deformable attention and residual structures within the fusion encoding module. Specifically, by developing a uniform sampling strategy, our method can easily sample from 2D image and 3D voxel features spontaneously, thus exploiting flexible adaptability and avoiding explicit transformation to the bird's eye view space during the feature concatenation process. Through extensive experiments on a popular autonomous driving benchmark dataset, nuScenes, our method achieves state-of-the-art single model performance of 72.6% mAP and 75.1% NDS in the 3D object detection task without test time augmentation.

## 1 Introduction

Autonomous driving technologies typically rely on multiple sensors for safety considerations, such as LiDAR (Chen et al., 2023; Yin et al., 2021; Wang et al., 2020; Lang et al., 2019), cameras (Wang et al., 2021b; 2022), and radar (Meyer & Kuschk, 2019; Meyer et al., 2021). These sensors possess distinct characteristics. For example, LiDAR can provide accurate yet sparse point clouds with 3D information, while images have dense features but lack such depth information. To enhance performance, multi-modal fusion can be used to integrate the strengths of these sensors. By combining information from multiple sensors, autonomous driving systems can achieve better accuracy and robustness, making them more reliable for real-world applications. Concatenating multi-modality features via simple concatenation in bird's eye view (BEV) space becomes a defacto standard to achieve state-of-the-art performance. As shown in Figure 1, current fusion framework fuses features from LiDAR point cloud and images in BEV space via simple concatenation (Liu et al., 2023; Liang et al., 2022) or a certain transformer architecture (Yan et al., 2023). However, we conjecture that these approaches has certain two limitations.

In order to fuse information at BEV level, we must first transform the 2D image features into 3D via certain geometry view transformation (Philion & Fidler, 2020). This process requires using a monocular depth estimation module which is an ill-posed problem and can generate inaccurate feature alignment. We believe that a superior approach is to exploit features from sparse point cloud to assist this process. Concurrently, Yan et al. (2023) proposes a transformer to leverage positional encoding to encode image features, which can be viewed as an alternative approach to alleviate this issue. However, all aforementioned methods explicitly transform the point voxel features into BEV space before the fusion module by compressing the Z-axis dimensional features into vectors. This may hinder the performance of downstream tasks that involves height information, such as 3D object detection where one needs to predict the height of the bounding box.

To tackle above problems, we propose a novel multimodal fusion framework for 3D object detection, dubbed FusionFormer to address these challenges. As shown in Figure 1 (c), FusionFormer can generate fused BEV features by sequentially fusing LiDAR and image features with deformable

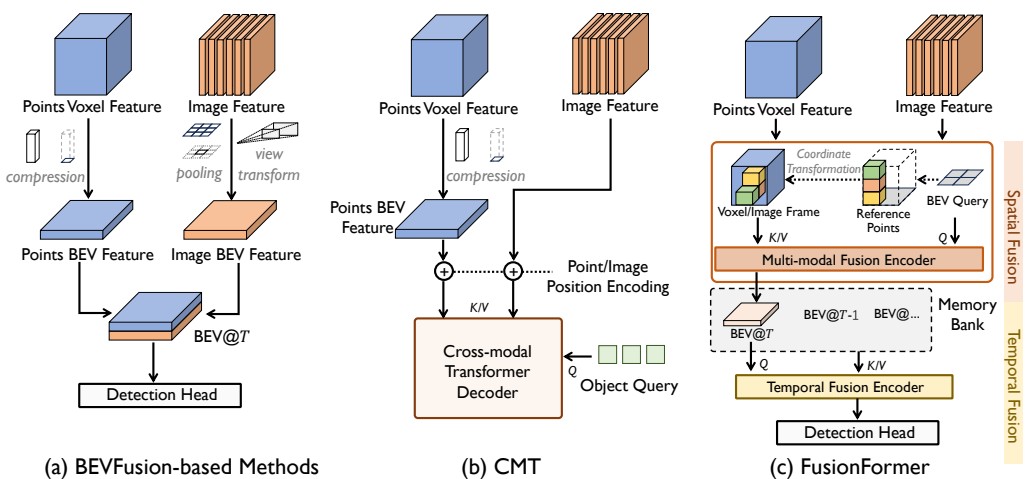

Figure 1: **Comparison between state-of-the-art methods and our FusionFormer.** **(a)** In BEVFusion-based methods, the camera features and points features are transformed into BEV space and fused with concatenation. **(b)** In CMT, the points voxel features are first compressed into BEV features. And the LiDAR BEV features and image features are encoded with the same positional encoding and tokenized as the Key and Value. Then, each object Query is passed into a transformer decoder to directly generate the prediction result. **(c)** In FusionFormer, the fusion of multi-modal features is achieved by sequentially interacting BEV queries with original point cloud voxel features and image features. This interaction leverages the depth references provided by point cloud features for the view transformer of image features, while the image features complement the sparsity of point cloud features. As a result, more accurate and dense fused BEV representations are obtained. Additionally, FusionFormer incorporates a temporal fusion encoding module, enabling the fusion of BEV features from historical frames.

attention (Zhu et al., 2020), which inherently samples features at the reference points corresponding to the BEV queries. By developing a uniform sampling strategy, our FusionFormer can easily sample from 2D image and 3D voxel features at the same time thus exhibits flexible adaptability across different modality inputs, and avoids explicit transformation and the need of monocular depth estimation. As a result, multi-modal features can be input in their original forms avoiding the information loss when transforming into BEV features. During the fusion encoding process, the point cloud features can serve as depth references for the view transform of image features, while the dense semantic features from images reciprocally complement the sparsity of point cloud features, leading to the generation of more accurate and dense fused BEV features. Notably, FusionFormer performs well even in the presence of missing point cloud or image features. We also propose a plug-and-play temporal fusion module along with our FusionFormer to support temporal fusion of BEV features from previous frames.

In addition, to verify the effectiveness and flexibility of our approaches, we use voxel features obtained from monocular depth estimation of only images to replace the features obtained from LiDAR point clouds to construct a FusionFormer that only uses camera modality.

In summary, we present the following contributions in this paper:

- We notice that state-of-the-art multi-modality frameworks need explicitly compressing the voxel features into BEV space before fusing with image features might lead to inferior performance, and propose a novel transformer based framework with a uniform sampling strategy to address this issue.

- We also demonstrate that our method is flexible and can be transformed into a camera only 3D object detector by replacing the LiDAR features to image features with monocular depth estimation.

- Our method achieves state-of-the-art single model performance of 72.6% mAP and 75.1% NDS in the 3D object detection task of the nuScenes dataset without test time augmentation.

## 2    RELATED WORK

**Visual-centric 3D Object Detection.**    In recent years, camera-based 3D object detection has gained increasing attention in the field of autonomous driving. Early approaches relied on predicting the 3D parameters of objects based on the results of 2D object detection (Park et al., 2021; Wang et al., 2021b). Recently, BEV-based 3D object detection has become a hot research topic (Xie et al., 2022). Compared to previous methods, BEV-based 3D object detection can directly output 3D object detection results around the vehicle using multi-view camera images, without requiring post-processing of detection results in overlapping regions. Inspired by LSS (Philion & Fidler, 2020), recent works like BEVDet (Huang et al., 2021) and BEVDepth (Li et al., 2023) have used bin-based depth prediction to transform multi-view camera features into BEV space. PETR (Liu et al., 2022a) achieves a camera-based BEV method with transformer by adding 3D position encoding. DETR3D (Wang et al., 2022) and BEVFormer (Li et al., 2022c) use deformable attention to make the query under BEV space interact with local features related to its position projection range during the transformer process, achieving the transformation from multi-view camera space to BEV space.

**LiDAR-centric 3D Object Detection.**    LiDAR-based 3D object detection methods can be categorized into different types based on the representation form of point cloud features. Point-wise methods extract features directly from individual points and output 3D object detection results end-to-end (Qi et al., 2018; Paigwar et al., 2019). BEV-based methods, on the other hand, construct intermediate feature forms before transforming them into BEV space (Yin et al., 2021). For instance, VoxelNet (Zhou & Tuzel, 2018) voxelizes the raw point cloud and applies sparse 3D convolutions to obtain voxel features. These features are subsequently compressed along the Z dimension to obtain BEV features. In contrast, Pointpillar (Lang et al., 2019) projects the point cloud into multiple pillars and pools the points within each pillar to extract features for BEV-based detection.

**Temporal-aware 3D Object Detection.**    Temporal fusion has emerged as a hot research topic in the field of 3D object detection for its ability to enhance detection stability and perception of target motion. BEVFormer (Li et al., 2022c) uses spatiotemporal attention to fuse the historical BEV features of the previous frame with current image features. BEVDet4D (Huang & Huang, 2022) employs concatenation to fuse temporally aligned BEV features from adjacent frames. SOLOFusion (Park et al., 2022) further leverages this approach to achieve long-term temporal fusion. Some methods perform temporal information fusion directly on the original feature sequences based on query. For instance, PETRv2 (Liu et al., 2022b) employs global attention and temporal position encoding to fuse temporal information, while Sparse4D (Lin et al., 2022) models the relationship between multiple frames based on sparse attention. Additionally, StreamPETR (Wang et al., 2023b) introduces a method for long-term fusion by leveraging object queries from past frames.

**Multi-modal 3D Object Detection.**    Fusing multi-sensory features becomes a de-facto standard in 3D perception tasks. BEVFusion-based methods (Liu et al., 2023; Liang et al., 2022; Cai et al., 2023) obtain image BEV features using view transform (Philion & Fidler, 2020; Li et al., 2022c) and concatenates them with LIDAR BEV features via simple concatenation. However, such simple stragety may fail to fully exploit the complementary information between multi-modal features. Another line of approaches construct transformer (Bai et al., 2022; Wang et al., 2023a; Yang et al., 2022) based architectures to perform interaction between image and point-cloud features. These methods relies simultaneously on both image and point cloud modal features, which presents challenges in cases of robustness scenarios when missing a modality data. Concurrently, Yan et al. (2023) proposes a method, dubbed CMT, which adopts 3D position encoding to achieve end-to-end multimodal fusion-based 3D object detection using transformer. Nonetheless, the aforementioned fusion methods rely on compressing point cloud voxel features into BEV representations, which can result in the loss of the height information. To tackle this, UVTR (Li et al., 2022b) introduced knowledge transfer to perform voxel-level multi-modal fusion by directly combining LiDAR voxel features with image voxel features obtained through LSS. However, this approach did not yield notable improvements in performance. Unlike these approaches, FusionFormer demonstrates enhanced adaptability to the input format of multimodal features, allowing direct utilization of point cloud features in voxel form. Moreover, by incorporating deformable attention and residual structures within the fusion encoding module, FusionFormer can achieve both multimodal feature complementarity and robustness in handling missing modal data.

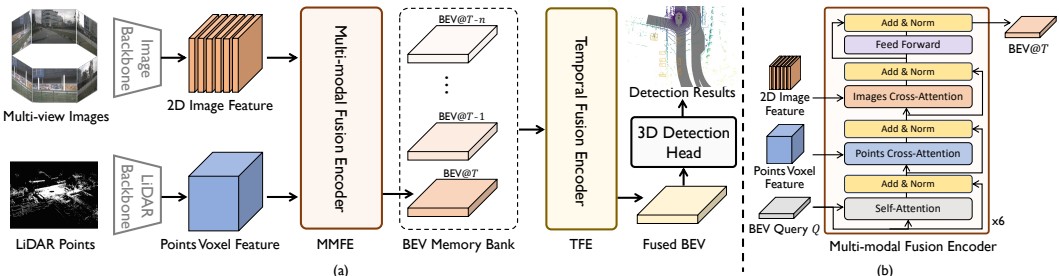

Figure 2: **(a) The framework of the FusionFormer.** The LiDAR point cloud and multi-view images are processed separately in their respective backbone networks to extract voxel features and image features. These features are then inputted into a multi-modal fusion encoder (MMFE) to generate the fused BEV features. The fused BEV features of the current frame, along with the BEV features from historical frames, are jointly fed into a temporal fusion encoder (TFE) to obtain the multi-modal temporal fused BEV features. Finally, the features are utilized in the detection head to produce the final 3D object detection results. **(b) The architecture of the Multi-modal Fusion Encoder (MMFE).** The BEV queries are initialized and subsequently subjected to self-attention. They are then sequentially utilized for cross-attention with the point cloud voxel features and image features. The resulting BEV queries, updated through a feed-forward network, are propagated as inputs to the subsequent encoder layers. Following multiple layers of fusion encoding, the ultimate fused BEV feature is obtained.

## 3 METHOD

Here we present our method in detail. Figure 2 (a) illustrates our proposed FusionFormer for multimodal temporal fusion. By utilizing a fusion encoder based on deformable attention (Lin et al., 2022), LiDAR and image features are transformed into fused BEV features. Compared to previous approaches such as BEVFusion (Liu et al., 2023; Liang et al., 2022), FusionFormer can adapt to different feature representations of different modalities without requiring pre-transformation into BEV space. The image branch can retain its original 2D feature representation, while the point cloud branch can be represented as BEV features or voxel features. Detailed information regarding the image branch and point cloud branch can be found in the A.1 section of the appendix. The temporal fusion module utilizes deformable attention to fuse BEV features from the current and previous frames that have been temporally aligned. Then the processed multimodal temporal fusion BEV features are input into the detection task head to obtain 3D object detection results.

### 3.1 MULTI-MODAL FUSION ENCODER

As illustrated in Figure 2 (b), the fusion encoding module consists of 6 layers, each incorporating self-attention, points cross-attention, and images cross-attention. In accordance with the standard transformer architecture, the BEV queries are subjected to self-attention following initialization. Subsequently, points cross-attention is executed to facilitate the integration of LiDAR features, which is further enhanced through images cross-attention to fuse image features. The encoding layer outputs the updated queries as input to the next layer after being processed through a feed-forward network. After 6 layers of fusion encoding, the final multimodal fusion BEV features are obtained.

**BEV Queries.** We partition the BEV space within the surrounding region of interest (ROI) range around the vehicle's center into a grid of $H \times W$ cells. Correspondingly, we define a set of learnable parameters $Q$ to serve as the queries for the BEV space. Each $q$ corresponds to a cell in the BEV space. Prior to inputting Q into the fusion encoder, the BEV queries are subjected to position encoding based on their corresponding BEV spatial coordinates (Li et al., 2022c).

**Self-Attention.** To reduce computational resource usage, we implemented the self-attention based on deformable attention. Each BEV query interacts only with its corresponding queries within the ROI range. This process is achieved through feature sampling at the 2D reference points for each

query as illustrated below:

$$SA(Q_p) = DefAttn(Q_p, p, Q) \tag{1}$$

where $Q_p$ represents the BEV query at point $p = (x, y)$.

**Points Cross-Attention.** The points cross-attention layer is also implemented based on deformable attention, but the specific manner in which points cross-attention is implemented varies depending on the form of the LiDAR points features. For the case where BEV features are used as input, we implement the points cross-attention layer as follows:

$$PCA_{2D}(Q_p, B_{pts}) = DefAttn(Q_p, P_{2D}, B_{pts}) \tag{2}$$

where $B_{pts}$ represents the BEV features output by the LiDAR branch, and $P_{2D} = (x_{2D}, y_{2D})$ represents the 2D projection of the coordinate $p = (x, y)$ onto the point cloud BEV space.

For the case where voxel features are used as input, the points cross-attention layer is implemented as follows:

$$PCA_{3D}(Q_p, V_{pts}) = \sum_{i=1}^{N_{ref}} DefAttn(Q_p, P_{3D}(p, i), V_{pts}) \tag{3}$$

where $V_{pts}$ represents the voxel features output by the LiDAR branch.

To obtain the 3D reference points, we first expand the grid cell corresponding to each BEV query with a height dimension, similar to the pillar representation (Lang et al., 2019). Then, from each pillar corresponding to a query, we sample a fixed number of $N_{ref}$ reference points, which are projected onto the point cloud voxel space using the projection equation $P_{3D}$. Specifically, for each query located at $p = (x, y)$, a set of height anchors $\{z_i\}_{i=1}^{N_{ref}}$ are defined along its $Z$-axis. Consequently, for each BEV query $Q_p$, a corresponding set of 3D reference points $(x, y, z_i)_{i=1}^{N_{ref}}$ is obtained. And the projection equation is as follow:

$$P_{3D}(p, i) = (x_{pts}, y_{pts}, z_{pts}) \tag{4}$$

where $P_{3D}(p, i)$ is the projection of the i-th 3D reference point of BEV query $Q_p$ in the LiDAR space.

**Images Cross-Attention.** The implementation of the images cross-attention is similar to the points cross-attention with voxel features as input. Since the images have multi views, the 3D reference points of each query can only be projected onto a subset of the camera views. Following BEV-Former (Li et al., 2022c), we denote the views that can be projected as $V_{hit}$. Therefore, the images cross-attention process can be expressed as:

$$ICA(Q_p, F) = \frac{1}{V_{hit}} \sum_{i=1}^{N_{ref}} \sum_{j=1}^{V_{hit}} DefAttn(Q_p, P(p, i, j), F_j) \tag{5}$$

where $j$ is the index of the camera view, $F_j$ represents the image features of the $j$-th camera, and $P(p, i, j)$ represents the projection point of the $i$-th 3D reference point $(x, y, z_i)$ of query $Q_p$ in the image coordinate system of the $j$-th camera.

### 3.2 TEMPORAL FUSION ENCODER

As shown in Figure 3, the temporal fusion encoder (TFE) consists of three layers, each comprising BEV temporal-attention and feedforward networks. At the first layer, the queries are initialized with the BEV features of the current frame and then updated through temporal-attention using historical BEV features. The resulting queries are passed through a feedforward network and serve as input to the next layer. After three layers of fusion encoding, the final temporal fusion BEV features are obtained. The temporal-attention process can be expressed as:

$$TCA(Q_p, B) = \sum_{i=0}^{T} DefAttn(Q_p, P, B_{t-i}) \tag{6}$$

where $B_{t-i}$ represents the BEV feature at time $t - i$.

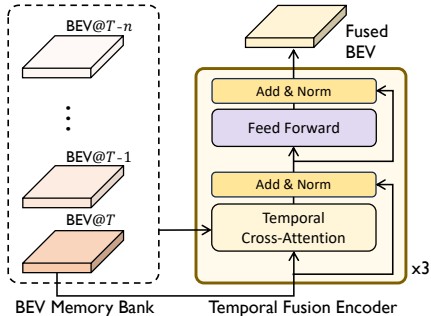

Figure 3: **Temporal Fusion Encoder (TFE).** The initial set of BEV queries is formed by utilizing the BEV features of the current frame. These queries are then subjected to cross-attention with historical BEV features, including the current frame. The resulting queries are updated through a feed-forward network and serve as inputs for the subsequent layer. Through multiple layers of temporal fusion encoding, the final output is obtained, representing the temporally fused BEV feature.

Figure 4: **Fusion with depth prediction.** After being processed by the backbone network, the multi-view image features are split into two branches. One branch utilizes a feature pyramid network (FPN) to extract multi-scale image features. The other branch employs a monocular depth prediction network to estimate depth and utilizes 3D convolution to encode the depth predictions. The multi-scale image features and the depth embedding are jointly input into the encoder to obtain the BEV features.

## 3.3 FUSION WITH DEPTH PREDICTION

The flexibility of FusionFormer enables us to approximate the point cloud branch in scenarios where only camera images are available by adding an image-based monocular depth prediction branch. As illustrated in Figure 4, we propose a depth prediction network to generate interval-based depth predictions from input image features. 3D convolution is utilized to encode the depth prediction results as voxel features in each camera frustum. Depth cross-attention is then employed to fuse the depth features. The process of depth cross-attention is defined as follows:

$$DCA(Q_p, D) = \frac{1}{V_{hit}} \sum_{i=1}^{N_{ref}} \sum_{j=1}^{V_{hit}} DefAttn(Q_p, P(p, i, j), D_j) \tag{7}$$

where $D_j$ denotes the encoded depth prediction features of the j-th camera, and $P(p, i, j)$ represents the projection point of the i-th 3D reference point $(x, y, z_i)$ of query $Q_p$ onto the frustum coordinate system of the j-th camera.

## 4 EXPERIMENTS

This section presents the performance of our proposed FusionFormer on the task of 3D object detection, along with several ablation studies that analyze the benefits of each module in our framework.

### 4.1 EXPERIMENTAL SETUPS

**Datasets and metrics.** We conducted experiments on the nuScenes dataset (Caesar et al., 2020) to evaluate the performance of our proposed method for 3D object detection in autonomous driving. The nuScenes dataset consists of 1.4 million 3D detection boxes from 10 different categories, with each frame of data containing 6 surround-view camera images and LiDAR point cloud data. We employ the nuScenes detection metrics NDS and mAP as evaluation metrics for our experiments.

**Implementation details.** We conducted algorithmic experiments using the open-source project MMDetection3D (Contributors, 2020) based on PyTorch. Specifically, we selected VoVNet-99 (Lee & Park, 2020) as the backbone for the image branch, generating multi-scale image features through

Table 1: Performance comparison on the nuScenes test set. "L" is LiDAR. "C" is camera. "T" is temporal. The results are evaluated using a single model without any test-time-augmentation or ensembling techniques.

| Methods | Modality | NDS↑ | mAP↑ | mATE↓ | mASE↓ | mAOE↓ | mAVE↓ | mAAE↓ |
|---|---|---|---|---|---|---|---|---|
| PointPainting(Vora et al.) | CL | 61.0 | 54.1 | 38.0 | 26.0 | 54.1 | 29.3 | 13.1 |
| PointAugmenting(Wang et al.) | CL | 71.1 | 66.8 | 25.3 | 23.5 | 35.4 | 26.6 | 12.3 |
| MVP(Chen et al.) | CL | 70.5 | 66.4 | 26.3 | 23.8 | 32.1 | 31.3 | 13.4 |
| FusionPainting(Xu et al.) | CL | 71.6 | 68.1 | 25.6 | 23.6 | 34.6 | 27.4 | 13.2 |
| TransFusion(Bai et al.) | CL | 71.7 | 68.9 | 25.9 | 24.3 | 35.9 | 28.8 | 12.7 |
| BEVFusion(Liu et al.) | CL | 72.9 | 70.2 | 26.1 | 23.9 | 32.9 | 26.0 | 13.4 |
| BEVFusion(Liang et al.) | CL | 73.3 | 71.3 | **25.0** | 24.0 | 35.9 | 25.4 | 13.2 |
| UVTR(Li et al.) | CL | 71.1 | 67.1 | 30.6 | 24.5 | 35.1 | **22.5** | 12.4 |
| CMT(Yan et al.) | CL | 74.1 | 72.0 | 27.9 | 23.5 | 30.8 | 25.9 | 11.2 |
| DeepInteraction(Yang et al.) | CL | 73.4 | 70.8 | 25.7 | 24.0 | 32.5 | 24.5 | 12.8 |
| BEVFusion4D-S(Cai et al.) | CL | 73.7 | 71.9 | - | - | - | - | - |
| BEVFusion4D(Cai et al.) | CLT | 74.7 | **73.3** | - | - | - | - | - |
| FusionFormer-S | CL | 73.8 | 70.8 | 26.7 | **23.4** | 28.9 | 25.8 | 10.7 |
| FusionFormer | CLT | **75.1** | 72.6 | 26.7 | 23.6 | **28.6** | 22.5 | **10.5** |

Table 2: Performance comparison on the nuScenes val set. "L" is LiDAR. "C" is camera. "T" is temporal. The "-S" indicates that the model only utilizes single-frame BEV features without incorporating temporal fusion techniques. The results are evaluated using a single model without any test-time-augmentation or ensembling techniques.

| Methods | Image Backbone | LiDAR Backbone | Modality | mAP↑ | NDS↑ |
|---|---|---|---|---|---|
| TransFusion(Bai et al.) | DLA34 | voxel0075 | CL | 67.5 | 71.3 |
| BEVFusion(Liu et al.) | Swin-T | voxel0075 | CL | 68.5 | 71.4 |
| BEVFusion(Liang et al.) | Swin-T | voxel0075 | CL | 67.9 | 71.0 |
| UVTR(Li et al.) | R101 | voxel0075 | CL | 65.4 | 70.2 |
| CMT(Yan et al.) | VoV-99 | voxel0075 | CL | 70.3 | 72.9 |
| DeepInteraction(Yang et al.) | R50 | voxel0075 | CL | 69.9 | 72.6 |
| BEVFusion4D-S(Cai et al.) | Swin-T | voxel0075 | CL | 70.9 | 72.9 |
| BEVFusion4D(Cai et al.) | Swin-T | voxel0075 | CLT | **72.0** | 73.5 |
| FusionFormer-S | VoV-99 | voxel0075 | CL | 70.0 | 73.2 |
| FusionFormer | VoV-99 | voxel0075 | CLT | 71.4 | **74.1** |

FPN (Lin et al., 2017). The input image size was set to $1600 \times 640$. For the LiDAR point cloud branch, VoxelNet (Zhou & Tuzel, 2018) was used as the backbone. The input LiDAR point cloud was voxelized with a size of $0.075m$. The size of the BEV queries was set to $200 \times 200$. During the training process, we loaded the pre-trained weights of the image branch backbone on Fcos3D (Wang et al., 2021b). The point cloud branch did not require pre-trained weights and was directly trained end-to-end with the model. We present a 3D detection head based on Deformable DETR (Zhu et al., 2020) that outputs 3D detection boxes and velocity predictions directly from BEV features without the need for non-maximum suppressing. To address the unstable matching problem encounterined in DETR-like detection heads and accelerate training convergence, we applied the query denoising strategy (Li et al., 2022a) during the training process. The model was trained for 24 epochs with the class-balanced grouping and sampling (CBGS) strategy (Zhu et al., 2019).

## 4.2 COMPARISON WITH STATE-OF-THE-ART METHODS

As shown in Table 1, FusionFormer achieves 75.1% NDS and 72.6% mAP on the nuScenes test dataset for 3D object detection, outperforming state-of-the-art methods. We used a single model fused with 8 frames of historical BEV features without any test-time-augmentation or ensembling techniques. We also compared the performance of FusionFormer with other methods on the nuScenes val dataset as shown in Table 2. Our proposed FusionFormer achieves state-of-the-art performance on both single-frame and temporal fusion scenarios with NDS scores of 73.2% and 74.1%. Several detection results on the nuScenes test set of FusionFormer are shown in Figure 5.

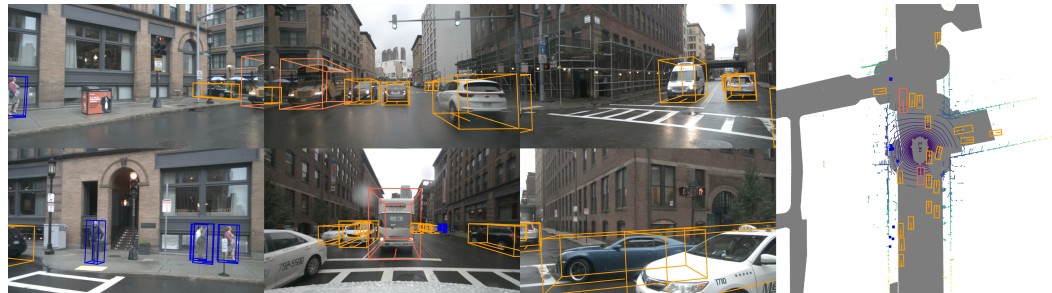

Figure 5: **Qualitative detection results in the nuScenes test set.** Bounding boxes with different colors represent Cars(•), Pedestrians(•), Bus(•) and Truck(•).

Table 3: Results of camera based 3D detection fused with depth prediction.

| Method | mAP↑ | NDS↑ |
|---|---|---|
| BEVFormer(Li et al.) | 41.6 | 51.7 |
| FusionFormer-Depth | **43.9** | **53.3** |

Table 4: Robustness performance on the nuScenes val set. "L" is LiDAR. "C" is camera.

| Method | Modality | mAP↑ | NDS↑ |
|---|---|---|---|
| FusionFormer | C | 34.3 | 45.5 |
| FusionFormer | L | 62.5 | 68.6 |
| FusionFormer | CL | 71.4 | 74.1 |

### 4.3 CAMERA BASED 3D DETECTION FUSED WITH DEPTH PREDICTION

As shown in Table 3, FusionFormer achieves $53.3\%$ NDS and $43.9\%$ mAP on the nuScenes val dataset with only camera images input by fused with the depth prediction results. Compared with the baseline BEVFormer, the NDS and mAP increased by $1.6\%$ and $2.3\%$ respectively. In particular, we found that after introducing the depth prediction branch, the BEV features output by the encoder can converge better. This may be because the depth information carried by the depth prediction branch allows the model to focus more accurately on the target location. As shown in Figure 6 (a), compared to BEVFormer, the BEV features obtained through FusionFormer-Depth are noticeably more focused on the target location.

### 4.4 ROBUSTNESS STUDY

During the training process, we incorporated modality mask (Yan et al., 2023; Yu et al., 2023) to enhance the model's robustness to missing modality data. As demonstrated in Table 4, our model can produce desirable results even in scenarios where image or point cloud data is missing, showcasing its strong robustness. These findings highlight the potential of our approach for addressing challenges in multi-modal learning and its potential for practical real-world applications.

### 4.5 ABLATION STUDY

In this section, we investigate the influence of each module on the performance of our proposed multi-modal fusion model for 3D detection. We adopted ResNet-50 (He et al., 2016) as the backbone for the image branch, with an input resolution of $800\times320$ for the image and a voxel size of $0.1m$ for the point cloud branch, outputting $150\times150$ BEV features. It is noteworthy that, all the experiments presented in this section were based on single frame without incorporating temporal fusion techniques. The models were trained for 24 epochs without utilizing the CBGS (Zhu et al., 2019) strategy.

**LiDAR Features.** In order to evaluate the impact of fusing voxel features from point cloud, we conducted experiments by comparing the model's performance with LiDAR features using BEV and voxel representations. Table 5 presents the results of all models. In contrast to inputting LiDAR features in the form of BEV, the use of voxel input format leads to superior model performance. Notably, the prediction errors for object center location and orientation are significantly reduced. This may be attributed to the preservation of more object structural information of the Z-axis in the voxel format, resulting in more accurate detection outcomes.

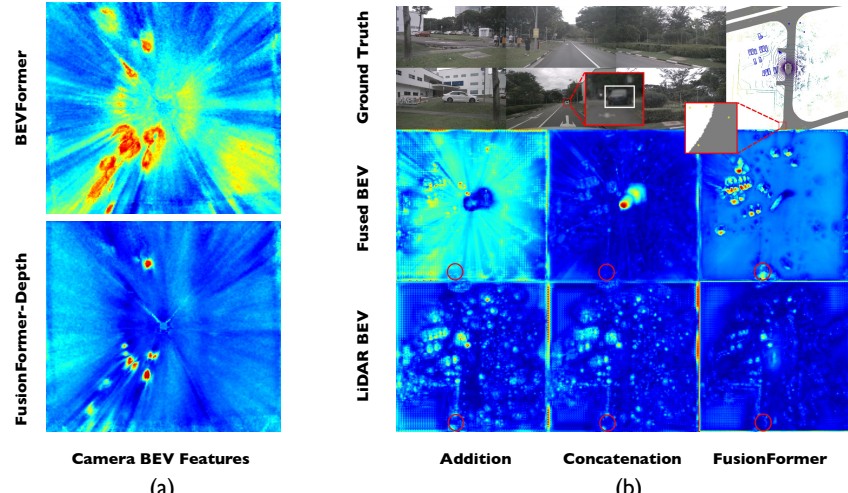

Figure 6: **(a) Visualization of the camera BEV features of BEVFormer and FusionFormer-Depth.** The BEV features obtained through FusionFormer-Depth are noticeably more focused on the target location than BEVFormer. **(b) Illustrations of the fused BEV features and LiDAR BEV features of different fusion methods.** The car labeled in the image are not annotated in the ground truth because they are far away and the LiDAR captures fewer points. FusionFomer is capable of better integrating multimodal features and can detect distant objects using image information even when the point cloud is sparse.

Table 5: Study for the representation of the LiDAR feature on the nuScenes val set.

| LiDAR | mAP↑ | NDS↑ | mATE↓ | mAOE↓ |
|---|---|---|---|---|
| BEV | 61.3 | 66.1 | 35.7 | 36.9 |
| Voxel | **62.7** | **67.3** | **34.4** | **31.4** |

Table 6: Ablation study of the modality fusion module on the nuScenes val set.

| Fusion Method | mAP↑ | NDS↑ |
|---|---|---|
| Addition | 59.3 | 64.6 |
| concatenation | 59.2 | 64.5 |
| Ours | **62.7** | **67.3** |

**Modality Fusion.** We conducted a comparative analysis of our proposed modality fusion method with other fusion methods to evaluate their performance. In the case of the fusion methods of addition and concatenation, the image BEV features were obtained through BEVFormer(Li et al., 2022c). The experimental results are presented in Table 6. As shown in Figure 6 (b), compared to other fusion methods, the fused BEV features obtained through FusionFormer exhibit a stronger response to the targets. Specifically, the distant cars labeled in the image are excluded from the ground truth (GT) annotations due to the limited points captured by LiDAR. Consequently, conventional multimodal fusion methods, such as simple addition and concatenation, fail to effectively incorporate these distant objects. In contrast, our proposed method, FusionFormer, enables enhanced fusion of multimodal features. It leverages the complementary information from image data to detect distant objects even in scenarios with sparse point cloud data.

## 5 CONCLUSION

In this paper, we propose a novel transformer-based framework with a uniform sampling strategy that overcomes the limitations of existing multi-modality frameworks. Our approach eliminates the need for compressing voxel features into BEV space before fusion with image features, resulting in superior performance. We demonstrate the versatility of our method by transforming it into a camera-only 3D object detector, utilizing image features obtained through monocular depth estimation instead of LiDAR features. Our method achieves state-of-the-art performance in the 3D object detection task on the nuScenes dataset. In future, we will explore the applications of FusionFormer in other tasks, such as map segmentation.

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
