# OpenReview forum: "FusionFormer: A Multi-sensory Fusion in Bird's-Eye-View and Temporal Consistent Transformer for 3D Object Detection"
_ICLR.cc/2024/Conference — Submitted to ICLR 2024_

### Official Review · Reviewer_5zjx · 2023-10-17

**Soundness:** 3 good
**Presentation:** 3 good
**Contribution:** 2 fair
**Rating:** 6
**Confidence:** 2

**Summary:**

This paper proposes a multi-modal fusion transformer-based framework for 3D object detection. The idea of this paper is intuitive and this paper is easy to understand. The performance on nuScenes is good.

**Strengths:**

1.This paper is easy to read.
2.The performance of the model in this paper show superiority in nuScenes.
3.The idea is intuitive.

**Weaknesses:**

1.Only one dataset is used in this paper. The generality of the framework should be analyized.
2.In introduction, Fig.1(b) is not compared with their method in details.
3.The most  baselines only consider LiDAR and camera, which are not fair to compare. Can you compare with them without temporal information? And this can also further verify the superiority of  Multi-modal fusion encoder.
4. The two topics, i.e., multi-modal fusion and temporal consistence, are different, causing that the focus of this paper is unclear.

**Questions:**

Refer to the weakness.

---

> ### Author Response · Authors · 2023-11-20
> **Thanks for your comments!**
>
> Thank you for your constructive comments and suggestions, and they are exceedingly helpful for us to improve our paper. We have carefully incorporated them in the revised paper. In the following, your comments are first stated and then followed by our point-by-point responses.
> ## Q1 More experiments
> > Only one dataset is used in this paper. The generality of the framework should be analyized.
>
> **Author response:**
> It is worth noting that previous studies on multimodal fusion have primarily focused on the nuScenes dataset, and there is little research conducted on other dataset, such as Waymo. This may be attributed to the differences between the datasets since the Waymo dataset primarily consists of LiDAR data, and the available images predominantly capture a forward-facing view while lacking rearward images.
>
> We are trying to evaluate the generalizability of our method on the Waymo dataset and other tasks such as segmentation. The experiments are still ongoing. We will include relevant experimental results in the final version of our paper.
>
> ## Q2 Compared with CMT
> > In introduction, Fig.1(b) is not compared with their method in details.
>
> **Author response:**
> Thank you for your feedback. We have added a more detailed comparison in Figure 1(b).
>
> We also list the key differences with CMT here:
>
> - CMT **tokenizes** image and LiDAR features directly and uses a transformer decoder to generate object detection predictions, while our method generates **fused BEV features** that are then connected to the head for prediction.
> - CMT uses compressed **BEV lidar features**, while we preserve the Z-axis information by using **voxel features**.
> - CMT have challenges in achieving long term temporal fusion, while our method supports **long term temporal fusion**.
>
> ## Q3 Single frame BEV feature result
> > The most baselines only consider LiDAR and camera, which are not fair to compare. Can you compare with them without temporal information? And this can also further verify the superiority of Multi-modal fusion encoder.
>
> **Author response:**
> We present the results as requested on nuScenes test set. Note that we do not use any test-time augmentation. Our single-frame FusionFormer (FusionFormer-S) achieves competitive performance. Nonetheless, we emphasize we aim to propose a uniform architecture for multi-modal, temporal consistent architecture.
>
> *Performance comparison on the nuScenes test set. "L" is LiDAR. "C" is camera. "T" is temporal. The results are evaluated using a single model without any test-time-augmentation or ensembling techniques.*
> |Methods|Modality|NDS↑|mAP↑|mATE↓|mASE↓|mAOE↓|mAVE↓|mAAE↓|
> |:----:|:----:|:----:|:----:|:----:|:----:|:----:|:----:|:----:|
> |PointPainting|CL|61.0|54.1|38.0|26.0|54.1|29.3|13.1|
> |PointAugmenting|CL|71.1|66.8|25.3|23.5|35.4|26.6|12.3|
> |MVP|CL|70.5|66.4|26.3|23.8|32.1|31.3|13.4|
> |FusionPainting|CL|71.6|68.1|25.6|23.6|34.6|27.4|13.2|
> |TransFusion|CL|71.7|68.9|25.9|24.3|35.9|28.8|12.7|
> |BEVFusion|CL|72.9|70.2|26.1|23.9|32.9|26.0|13.4|
> |BEVFusion|CL|73.3|71.3|**25.0**|24.0|35.9|25.4|13.2|
> |UVTR|CL|71.1|67.1|30.6|24.5|35.1|**22.5**|12.4|
> |CMT|CL|74.1|72.0|27.9|23.5|30.8|25.9|11.2|
> |DeepInteraction|CL|73.4|70.8|25.7|24.0|32.5|24.5|12.8|
> |BEVFusion4D-S|CL|73.7|71.9| - | - | - | - | - |
> |BEVFusion4D|CLT|74.7|**73.3**| - | - | - | - | - |
> |FusionFormer-S|CL|73.8|70.8|26.7|**23.4**|28.9|25.8|10.7|
> |FusionFormer|CLT|**75.1**|72.6|26.7|23.6|**28.6**|**22.5**|**10.5**|
>
>
> ## Q4 Multi-modal and temporal fusion
> > The two topics, i.e., multi-modal fusion and temporal consistence, are different, causing that the focus of this paper is unclear.
>
> **Author response:**
> Thank you for your comments. Both multimodal fusion and temporal fusion are currently hot topics in the field of autonomous driving research, with significant implications for improving perception accuracy and stability. This study aims to present a unified Transformer-based multimodal temporal fusion framework. Our proposed Transformer-based multimodal fusion encoding module  adapts to various multimodal feature inputs, resulting in fused BEV features. Our approach outperforms fusion methods such as BEVFusion in terms of multimodal fusion performance. Furthermore, compared to previous transformer-based methods that do not support temporal fusion such as CMT and DeepInteraction, our approach, based on the proposed temporal fusion module, is capable of supporting temporal fusion and achieves state-of-the-art performance on the nuScenes dataset.
> In conclusion, we introduces a unified Transformer-based multimodal temporal fusion framework that achieves state-of-the-art results on the nuScenes dataset.

---

### Official Review · Reviewer_bQU3 · 2023-10-28

**Soundness:** 2 fair
**Presentation:** 3 good
**Contribution:** 2 fair
**Rating:** 6
**Confidence:** 5

**Summary:**

This work proposes a new spatial-temporal multi-modal fusion framework for 3D object detection. The proposed framework leverages 2D image features and 3D voxel features to generate BEV features and refine the features with temporal memory banks, which are then fed to the detection head to generate 3D object predictions. The method achieves leading performance on the nuScenes dataset.

**Strengths:**

* Overall the paper is well-written with clear technical pipeline.
* The proposed framework can work even when missing modality inputs, showing better robustness.
* The proposed method archives new SOTA results.

**Weaknesses:**

* The motivation is not clear or sufficient. The author claims that " state-of-the-art multi-modality frameworks need explicitly compressing the voxel features into BEV space" and the proposed approach is aimed to address this issue. However, existing works like DeepInteraction maintain per-modality representations (2D perspective for camera and voxel space for LiDAR) and learn the interactions between these cross-modality interactions without the need for BEV intermediate representations.  What's the difference between the proposed approach and DeepInteraction? Please fully discuss the differences and highlight the novelty of the proposed methods with SOTA methods.
* Lack of novelty. The proposed framework has a large overlay with both DeepInteraction and Bevformer. The BEV grid representation, temporal BEV feature fusion, and deformable attention have already been used in Bevformer and the cross-modality interaction of 2D features and 3D features have also been proposed in DeepInteractions. I would encourage the author to highlight the novelty and differences.
* The memory banks contain all the previous BEV features, which is very time-consuming for learning the interactions of current BEV feature maps and previous ones. Why not choose a recurrent-style temporal fusion mechanism?
* In the abstract and introduction, the author claims one of the main contributions is the residual architecture. However, in the method section, it is rarely mentioned.
* When comparing with other SOTAs, is temporal information used for all the other methods for a fair comparison?

**Questions:**

* The BEV visualization in Figure 6 (b) of the proposed method looks very good. Is this the result of a pure camera branch or generated from both LiDAR and camera inputs? What is the feature visualization of LiDAR BEV feature map?  How are the visual improvements after adding the camera modality?

---

> ### Author Response · Authors · 2023-11-20
> **Thanks for your comments!**
>
> Thank you for your constructive comments and suggestions, and they are exceedingly helpful for us to improve our paper. We have carefully incorporated them in the revised paper. In the following, your comments are first stated and then followed by our point-by-point responses.
> ## Q1 Differences with DeepInteraction
> > The motivation is not clear or sufficient. The author claims that " state-of-the-art multi-modality frameworks need explicitly compressing the voxel features into BEV space" and the proposed approach is aimed to address this issue. However, existing works like DeepInteraction maintain per-modality representations (2D perspective for camera and voxel space for LiDAR) and learn the interactions between these cross-modality interactions without the need for BEV intermediate representations. What's the difference between the proposed approach and DeepInteraction? Please fully discuss the differences and highlight the novelty of the proposed methods with SOTA methods.
>
> **Author response:**
>
> We have clarified our novelty and the difference with SOTA methods in the general response. And we detail the response to your question here.
>
> We apologize for any misunderstanding, however, we again list the key differences with DeepInteraction here:
>
> - DeepInteraction focus on the **interaction** between multimodal features, while our method focuses on **modal fusion**.
> - DeepInteraction preserves the original form of each modality feature after interaction, while our method generates **fused features**.
> - DeepInteraction uses compressed **BEV lidar features**, while we preserve the Z-axis information by using **voxel features**.
> - DeepInteraction relies on both modal inputs simultaneously, while our method exhibits **robustness** in the presence of **missing modalities**.
> - DeepInteraction is not easily applicable for temporal fusion, while our method supports **temporal fusion**.

---

> ### Author Response · Authors · 2023-11-20
> **Q2 Clarification of the novelty**
>
> ## Q2 Clarification of the novelty
> > Lack of novelty. The proposed framework has a large overlay with both DeepInteraction and Bevformer. The BEV grid representation, temporal BEV feature fusion, and deformable attention have already been used in Bevformer and the cross-modality interaction of 2D features and 3D features have also been proposed in DeepInteractions. I would encourage the author to highlight the novelty and differences.
>
> **Author response:**
>
> We have clarified our novelty and the difference with SOTA methods in the general response. And we detail the response to your question here.
> We also take this opportunity to emphasize the main contributions of our paper here:
>
> - **We advocate using point cloud features in __voxel format__ as opposed to using compressed LiDAR BEV features along the Z-axis.** The recent approaches, BEVFusion, DeepInteraction and CMT, rely on LiDAR BEV features which is compressed along the Z-axis. Compared to fusing LiDAR features in the form of BEV features, our voxel feature fusion approach exhibits remarkable improvements in both object center position prediction and orientation prediction accuracy as shown in Table 5 in our paper (we provide the table here for your convenience).
>
> *Study for the representation of the LiDAR feature on the nuScenes val-set, as in Table.*
> |LiDAR|mAP↑|NDS↑|mATE↓|mAOE↓|
> |:----:|:----:|:----:|:----:|:----:|
> |BEV|61.3|66.1|35.7|36.9|
> |Voxel|**62.7**|**67.3**|**34.4**|**31.4**|
>
> - **Based on our voxel feature fusion approach, we have discovered that it is possible to enhance visual-based 3D object detection by replacing LiDAR features with voxel features generated from *monocular depth predictions*.** This novel idea provides a new direction for improving the performance of visual-based 3D object detection.
>
> *Results of camera based 3D detection on the nuScenes val set.*
> |Method|Backbone|mAP|NDS|
> |:----:|:----:|:----:|:----:|
> |BEVFormer|R101-DCN|41.6|51.7|
> |FusionFormer-Depth|R101-DCN|**43.9**|**53.5**|
>
> - **We have proposed a unified multimodal temporal fusion method based on Transformers, which has achieved a new state-of-the-art performance on the nuScenes dataset.**
>
> We apologize for any misunderstanding, however, we again list the key differences between our method with BEVFormer & DeepInteraction here:
> **The difference between our method with BEVFormer**
> - BEVFormer only supports **single-modality** visual input. In contrast, our method focuses on providing a novel **multimodal** fusion solution.
> - BEVFormer adopts a recurrent-based temporal fusion approach that does not support long-term temporal fusion. In contrast, our method efficiently achieves **long-term temporal fusion**.
>
> **The difference between our method DeepInteraction**
> - DeepInteraction focus on the **interaction** between multimodal features, while our method focuses on **modal fusion**.
> - DeepInteraction preserves the original form of each modality feature after interaction, while our method generates **fused features**.
> - DeepInteraction uses compressed **BEV lidar features**, while we preserve the Z-axis information by using **voxel features**.
> - DeepInteraction relies on both modal inputs simultaneously, while our method exhibits **robustness** in the presence of **missing modalities**.
> - DeepInteraction is not easily applicable for temporal fusion, while our method supports **temporal fusion**.

---

> > ### Author Response · Authors · 2023-11-20
> > **Q3 Temporal fusion mechanism**
> >
> > ## Q3 Temporal fusion mechanism
> > > The memory banks contain all the previous BEV features, which is very time-consuming for learning the interactions of current BEV feature maps and previous ones. Why not choose a recurrent-style temporal fusion mechanism?
> >
> > **Author response:**
> > Regarding the choice of temporal fusion strategy, we opted for a sliding time window approach instead of a recurrent sequential fusion mechanism for several compelling reasons.
> > Currently, there are two prominent recurrent fusion mechanisms in the literature.
> > The first type, exemplified by BEVFormer, employs a fixed history frames fusion approach. However, extensive experiments with BEVFormer have demonstrated diminishing returns when fusing more than 5 frames during training. This observation suggests that there may be a limitation associated with forgetting historical frames. In contrast, our sliding time window strategy continues to deliver performance gains even when fusing up to 8 frames.
> >
> > *The Performance of BEVFormer on nuScenes val set with different frame numbers during training.*
> > |Frame|mAP↑|NDS↑|
> > |:----:|:----:|:----:|
> > |1|37.5|44.8|
> > |2|38.8|49.0|
> > |3|41.0|51.0|
> > |4|41.6|51.7|
> > |5|41.2|51.7|
> >
> > *The Study of the Performance of FusionFormer on nuScenes val set with different frame numbers during training.*
> > |Frame|mAP↑|NDS↑|
> > |:----:|:----:|:----:|
> > |1|66.5|70.4|
> > |2|67.8|71.2|
> > |4|68.2|71.5|
> > |8|68.6|71.7|
> >
> >
> > The second type of recurrent fusion approach, represented by VideoBEV, is a stream-based fusion method. However, implementing this approach poses challenges for incorporating crucial data augmentation techniques commonly used for point clouds, such as rotation, translation, and scaling. These techniques significantly impact the performance of points cloud based methods.
> > Furthermore, our designed temporal cross-attention mechanism enables simultaneous interactions between all historical BEV features and the current frame BEV features, ensuring efficient computation without compromising performance.
> >
> > |Latency (ms)|Camera Backbone|LiDAR Backbone|Multi-Modal Fusion Encoder|Temporal Fusion Encoder|Head|FPS|
> > |:----:|:----:|:----:|:----:|:----:|:----:|:----:|
> > |FusionFormer|20|124|79|22|23|3.8|
> >
> > Considering the aforementioned factors, we ultimately chose the sliding time window strategy for temporal fusion.

---

> > > ### Author Response · Authors · 2023-11-20
> > > **Q4 Residual architecture**
> > >
> > > ## Q4 Residual architecture
> > > > In the abstract and introduction, the author claims one of the main contributions is the residual architecture. However, in the method section, it is rarely mentioned.
> > >
> > > **Author response:**
> > > It appears that our initial statement may have caused some confusion. We have revised our statement to avoid any potential misinterpretation. We present the relevant statements from the original text regarding the residual structure:
> > > > To this end, we propose a novel end-to-end multi-modal fusion transformer-based framework, dubbed FusionFormer, that incorporates deformable attention and **residual structures** within the fusion encoding module.
> > >
> > > > We further implement a **residual structure** in our feature encoder to ensure the model’s ro- bustness in case of missing an input modality.
> > >
> > > > Notably, the multi-modal fusion encoder incorporates **residual structures**, ensuring the model's robustness in the presence of missing point cloud or image features.
> > >
> > > We would like to clarify that the residual structure is not a primary contribution of our work. The mention of the residual structure was intended to explain how our model performs well even in the absence of one modality, such as image or point cloud data.
> > > The residual structure is a widely used technique in deep learning, and we leverage it in our multimodal fusion module. Additionally, by incorporating a stochastic modality dropout strategy during training, our model demonstrates robust performance even in scenarios where one modality is missing, without suffering a complete failure. This feature makes our approach especially suitable for real-world autonomous driving settings where sensor dropout and modality data absence can occur.
> > >
> > > *Robustness performance on the nuScenes val set. ”L” is LiDAR. ”C” is camera.*
> > > |Method|Modality|mAP↑|NDS↑|
> > > |:----:|:----:|:----:|:----:|
> > > |FusionFormer|C|34.3|45.5|
> > > |FusionFormer|L|62.5|68.6|
> > > |FusionFormer|CL|71.4|74.1|

---

> > > > ### Author Response · Authors · 2023-11-20
> > > > **Q5 Single frame BEV feature result**
> > > >
> > > > ## Q5 Single frame BEV feature result
> > > > > When comparing with other SOTAs, is temporal information used for all the other methods for a fair comparison?
> > > >
> > > > **Author response:**
> > > > When comparing with other state-of-the-art (SOTA) methods in Tables 1 and 2, we have indicated the use of temporal information by adding a "T" symbol in the Modality column. In Table 2, we have included results for FusionFormer with and without the utilization of temporal information on the val set of nuScenes. Our approach achieves SOTA performance in terms of the NDS metric in both cases.
> > > > In the previous version of Table 1, we only presented the results of FusionFormer when incorporating temporal information. In the updated version, we have included an additional evaluation experiments for FusionFormer using single-frame BEV features on the nuScenes test set. The experimental results demonstrate that FusionFormer-S achieves competitive results on the nuScenes test set even without incorporating temporal information. We would like to emphasize that during the evaluation process, FusionFormer-S only utilizes a single model and does not employ any testing augmentation techniques.
> > > >
> > > > *Performance comparison on the nuScenes test set. "L" is LiDAR. "C" is camera. "T" is temporal. The results are evaluated using a single model without any test-time-augmentation or ensembling techniques.*
> > > > |Methods|Modality|NDS↑|mAP↑|mATE↓|mASE↓|mAOE↓|mAVE↓|mAAE↓|
> > > > |:----:|:----:|:----:|:----:|:----:|:----:|:----:|:----:|:----:|
> > > > |PointPainting|CL|61.0|54.1|38.0|26.0|54.1|29.3|13.1|
> > > > |PointAugmenting|CL|71.1|66.8|25.3|23.5|35.4|26.6|12.3|
> > > > |MVP|CL|70.5|66.4|26.3|23.8|32.1|31.3|13.4|
> > > > |FusionPainting|CL|71.6|68.1|25.6|23.6|34.6|27.4|13.2|
> > > > |TransFusion|CL|71.7|68.9|25.9|24.3|35.9|28.8|12.7|
> > > > |BEVFusion|CL|72.9|70.2|26.1|23.9|32.9|26.0|13.4|
> > > > |BEVFusion|CL|73.3|71.3|**25.0**|24.0|35.9|25.4|13.2|
> > > > |UVTR|CL|71.1|67.1|30.6|24.5|35.1|**22.5**|12.4|
> > > > |CMT|CL|74.1|72.0|27.9|23.5|30.8|25.9|11.2|
> > > > |DeepInteraction|CL|73.4|70.8|25.7|24.0|32.5|24.5|12.8|
> > > > |BEVFusion4D-S|CL|73.7|71.9| - | - | - | - | - |
> > > > |BEVFusion4D|CLT|74.7|**73.3**| - | - | - | - | - |
> > > > |FusionFormer-S|CL|73.8|70.8|26.7|**23.4**|28.9|25.8|10.7|
> > > > |FusionFormer|CLT|**75.1**|72.6|26.7|23.6|**28.6**|**22.5**|**10.5**|

---

> > > > > ### Author Response · Authors · 2023-11-20
> > > > > **Q6 BEV visualization**
> > > > >
> > > > > ## Q6 BEV visualization
> > > > > > The BEV visualization in Figure 6 (b) of the proposed method looks very good. Is this the result of a pure camera branch or generated from both LiDAR and camera inputs? What is the feature visualization of LiDAR BEV feature map? How are the visual improvements after adding the camera modality?
> > > > >
> > > > > **Author response:**
> > > > > Regarding Figure 6(b), the BEV feature represents the result of fusing image and LiDAR features. In our updated version, we have included visualizations of the LiDAR features.
> > > > > After incorporating image feature fusion in all three fusion methods, the responses to irrelevant objects such as trees and buildings in the point cloud feature are effectively suppressed. Notably, the fusion features obtained through FusionFormer exhibit even more significant suppression of irrelevant object responses. Moreover, the features related to vehicles and pedestrians appear denser, which can be attributed to the denser semantic information brought by the fusion of image features. These findings offer valuable insights into the enhanced performance of our proposed fusion method compared to existing approaches.

---

> ### Comment · Reviewer_bQU3 · 2023-11-21
>
> Thanks for the detailed feedback. I am still not convinced about the novelty part. From the current response, the voxel representation is novel and distinct with deepinteraction and bevformer. However, all the other argued novelty in comparisons are basically the differences between deepinteraction and bevformer which the proposed method builds upon. I would encourage the author to highlight the differences between the proposed method and deepinteraction&bevformer.

---

> > ### Author Response · Authors · 2023-11-22
> > **The differences between our method and DeepInteraction&BevFormer.**
> >
> > Thanks for your comments! We would like to confirm if it was our reply that caused the misunderstanding. What we previously listed in our response were not the differences between BEVFormer and DeepInteraction, but the differences between our method and BEVFormer&DeepInteraction. We apologize for any misunderstanding, however, we again list the key differences between our method and BEVFormer & DeepInteraction here:
> > **The difference between our method and BEVFormer**
> > - BEVFormer only supports **single-modality** visual input. In contrast, our method focuses on providing a novel **multimodal** fusion solution.
> > - BEVFormer adopts a recurrent-based temporal fusion approach that does not support long-term temporal fusion. In contrast, our method efficiently achieves **long-term temporal fusion**.
> >
> > **The difference between our method and DeepInteraction**
> > - DeepInteraction focus on the **interaction** between multimodal features, while our method focuses on **modal fusion**.
> > - DeepInteraction preserves the original form of each modality feature after interaction, while our method generates **fused features**.
> > - DeepInteraction uses compressed **BEV lidar features**, while we preserve the Z-axis information by using **voxel features**.
> > - DeepInteraction relies on both modal inputs simultaneously, while our method exhibits **robustness** in the presence of **missing modalities**.
> > - DeepInteraction is not easily applicable for temporal fusion, while our method supports **temporal fusion**.

---

> ### Comment · Reviewer_bQU3 · 2023-11-22
>
> Thanks for the response. Previously you are comparing proposed method vs Bevformer and then compare the proposed method vs Deepinteraction. However this is not sufficient for highlighting the novelties. I would encourage the author to list the differences between the proposed method vs (Bevformer+DeepInteraction). In other words, Bevformer and DeepInteraction is treated as a group. You should highlight the differences and novelties that neither BEVFormer nor DeepInteraction have. Otherwise, still it is unclear if the proposed method is just a combination of these two papers. For example, temporal fusion is not included in deepinteraction but is included in bevformer and if the temporal fusion is the same as the bevformer, then I think the technical contribution and novelty would be limited.

---

> > ### Author Response · Authors · 2023-11-22
> > **Clarification of the differences between our method and DeepInteraction&BevFormer**
> >
> > Thanks for your comments. It is important to clarify that our approach differs significantly from a mere combination of BEVFormer and DeepInteraction. We list the differences between our method and BEVFormer&DeepInteraction.
> >
> > - We propose a novel **multimodal fusion** method based on Transformer, which generates **unified moltimodal fusion features** from input multimodal data. Our approach differs significantly from a mere combination of BEVFormer and DeepInteraction. BEVFormer is a **visual single-modal** method that does not support multimodal feature input. On the other hand, DeepInteraction employs **modality interaction**, where point cloud and image features interact while maintaining separate representations, and **does not** generate **unified fusion features**. This is fundamentally different from our method, which is based on multimodal fusion as its core idea.
> >
> > - Our method utilizes **LIDAR voxel features** as input, enhancing the model's performance. DeepInteraction, however, uses **LiDAR BEV features** compressed in the Z-axis direction, while BEVFormer is limited to visual single-modal inputs and **lacks support for LiDAR features**.
> >
> > - Our method introduces an efficient **sliding time window-based long-term fusion** approach using Transformer. This approach differs from BEVFormer, which relies on **recurrent-style** temporal fusion. Furthermore, DeepInteraction, due to its interactive nature, lacks a unified fusion feature representation. To the best of our knowledge, a suitable application of long-term temporal fusion in the paradigm of DeepInteraction has not yet been established.
> >
> > - Our multimodal temporal fusion method demonstrates **robustness in the presence of missing modal inputs**. Even when one modality, such as image or point cloud, is missing, the model still achieves favorable results. In contrast, BEVFormer relies solely on a single image modality, rendering the model failure when all image data is missing. DeepInteraction's detection head depends on both image and point cloud modalities, leading to model failure if either one is absent.
> >
> > We believe these clarifications highlight the uniqueness and significance of our contributions and emphasize the differences between our method and BEVFormer&DeepInteraction.

---

### Official Review · Reviewer_q7W1 · 2023-10-29

**Soundness:** 3 good
**Presentation:** 3 good
**Contribution:** 2 fair
**Rating:** 3
**Confidence:** 4

**Summary:**

This work proposes a transformer-based framework for 3D multi-modality object detection. It mainly contains spatial fusion and temporal fusion modules to fuse cross-modality features and temporal features, respectively. Experiments prove the effectiveness of the proposed modules.

**Strengths:**

1. The direction of cross-modality fusion for 3D object detection is promising, which could bring potential effect to practical application.
2. The whole method is simple and easy to follow.
3. The presentation and writing is clear.

**Weaknesses:**

1. The core idea of utilizing BEV queries for temporal and cross-modality fusion is widely used in previous methods like BEVFormer. Although the proposed method is different in detailed design, the core application of BEV queries is unchanged. This harms the technical contribution of the proposed method.
2. The runtime comparisons are missing. Because this work incorporated several attention modules in different encoders, it's essential to report the latency of each module.
3. It's interesting that the proposed method can also support the camera-only setting in Figure 4. However, the performance in Table 3 seems not good enough compared with the clear BEV modeling like BEVDepth. Does it mean the fusion-based method in Figure 4 is not a good choice for the camera-only setting?

**Questions:**

Please refer to the weakness section.

---

> ### Author Response · Authors · 2023-11-20
> **Thanks for your comments!**
>
> Thank you for your constructive comments and suggestions, and they are exceedingly helpful for us to improve our paper. We have carefully incorporated them in the revised paper. In the following, your comments are first stated and then followed by our point-by-point responses.
> ## Q1 The technical contribution
> > The core idea of utilizing BEV queries for temporal and cross-modality fusion is widely used in previous methods like BEVFormer. Although the proposed method is different in detailed design, the core application of BEV queries is unchanged. This harms the technical contribution of the proposed method.
>
> **Author response:**
> We first clarify our novelty in the general response. And we detail the response to your question here.
>
> We apologize for any misunderstanding, however, we again list the key differences with BEVFormer here:
> - BEVFormer is a **visual only** method. As such, it is not possible for them to use BEV queries to fuse cross-modality information.
> - BEVFormer adopts a recurrent-based temporal fusion approach that does not support long-term temporal fusion. In contrast, our method efficiently achieves **long-term temporal fusion**.
>
> We also take this opportunity to emphasize the main contributions of our paper here:
>
> - **We advocate using point cloud features in __voxel format__ as opposed to using compressed LiDAR BEV features along the Z-axis.** The recent approaches, BEVFusion, DeepInteraction and CMT, rely on LiDAR BEV features which is compressed along the Z-axis. Compared to fusing LiDAR features in the form of BEV features, our voxel feature fusion approach exhibits remarkable improvements in both object center position prediction and orientation prediction accuracy as shown in Table 5 in our paper (we provide the table here for your convenience).
>
> *Study for the representation of the LiDAR feature on the nuScenes val-set, as in Table.*
> |LiDAR|mAP↑|NDS↑|mATE↓|mAOE↓|
> |:----:|:----:|:----:|:----:|:----:|
> |BEV|61.3|66.1|35.7|36.9|
> |Voxel|**62.7**|**67.3**|**34.4**|**31.4**|
>
> - **Based on our voxel feature fusion approach, we have discovered that it is possible to enhance visual-based 3D object detection by replacing LiDAR features with voxel features generated from *monocular depth predictions*.** This novel idea provides a new direction for improving the performance of visual-based 3D object detection.
>
> *Results of camera based 3D detection on the nuScenes val set.*
> |Method|Backbone|mAP|NDS|
> |:----:|:----:|:----:|:----:|
> |BEVFormer|R101-DCN|41.6|51.7|
> |FusionFormer-Depth|R101-DCN|**43.9**|**53.5**|
>
> - **We have proposed a unified multimodal temporal fusion method based on Transformers, which has achieved a new state-of-the-art performance on the nuScenes dataset.**
>
> ## Q2: Runtime comparison of module.
> > The runtime comparisons are missing. Because this work incorporated several attention modules in different encoders, it's essential to report the latency of each module.
>
> **Author response:**
> We have added a time analysis for different modules, and the results are presented in the table below on a single A100 GPU. The time for the multimodal fusion encoding module represents the total time for 6 layers of encoding, while the time for the temporal fusion encoding module represents the total time for 3 layers of encoding. The time for each attention module represents the time for a single module in a single layer of encoding.
> |Latency (ms)|Camera Backbone|LiDAR Backbone|Multi-Modal Fusion Encoder|Points Cross-Attention|Image Cross-Attention|Self-Attention|Temporal Fusion Encoder|Head|FPS|
> |:----:|:----:|:----:|:----:|:----:|:----:|:----:|:----:|:----:|:----:|
> |FusionFormer|20|124|79|1|5|6|22|23|3.8|
> |FusionFormer-S|20|124|80|1|5|6|-|23|4.0|
>
> Nonetheless, we already provide the runtime analysis against other baselines in Table 7 of Appendix. We replicate this table here for your convenience. Our method has a competitive runtime compared to those method.
>
> *Appendix Table 7 Efficiency comparison on the nuScenes val set. ”L” is LiDAR. ”C” is camera. ”T” is temporal. The ”-S” indicates that the model only utilizes single-frame BEV features without incorporating temporal fusion techniques.*
> |Methods|Modality|mAP|NDS|FPS|
> |:----:|:----:|:----:|:----:|:----:|
> |TransFusion|CL|67.5|71.3|3.2|
> |BEVFusion|CL|68.5|71.4|4.2|
> |UVTR|CL|65.4|70.2|2.6|
> |CMT|CL|70.3|72.9|**6.0**|
> |DeepInteraction|CL|69.8|72.6|1.7|
> |FusionFormer-S|CL|70.0|73.2|4.0|
> |FusionFormer|CLT|**71.4**|**74.1**|3.8|

---

> > ### Author Response · Authors · 2023-11-20
> > **Q3: Camera only results compared to BEVDepth.**
> >
> > ## Q3: Camera only results compared to BEVDepth.
> > > It's interesting that the proposed method can also support the camera-only setting in Figure 4. However, the performance in Table 3 seems not good enough compared with the clear BEV modeling like BEVDepth. Does it mean the fusion-based method in Figure 4 is not a good choice for the camera-only setting?
> >
> > **Author response:**
> > Thank you for your comments. Our research primarily focuses on exploring multimodal temporal fusion techniques for 3D object detection. The experimental setup described in Table 3 was specifically designed to demonstrate the flexibility of our approach.
> > By incorporating voxel features generated from monocular depth predictions instead of point cloud-based voxel features, our method significantly improves the performance of visual 3D object detection.
> > To ensure a fair comparison with the baseline method, we maintain consistent backbone architectures, BEV sizes, and other configurations with the baseline BEVFormer, while only introducing an additional depth prediction branch.
> > In contrast to BEVDepth, another method that integrates depth predictions, we achieve comparable results in terms of NDS on the nuScenes validation set without employing any data augmentation during training. Furthermore, our method outperforms BEVDepth in terms of mAP. These results highlight the flexibility of our approach and the benefits of fusing visual depth predictions for enhancing visual 3D object detection. The visualization of BEV features in Figure 6(a) clearly demonstrates that our approach significantly improves the model's ability to accurately perceive object distances, resulting in a more focused and precise localization of objects.
> >
> > *Results of camera based 3D detection on the nuScenes val set.*
> > |Method|Backbone|mAP|NDS|
> > |:----:|:----:|:----:|:----:|
> > |BEVFormer|R101-DCN|41.6|51.7|
> > |BEVDepth|R101-DCN|41.8|**53.8**|
> > |FusionFormer-Depth|R101-DCN|**43.9**|53.5|
> >
> > We believe that our framework provides a novel approach to improve the accuracy of distance perception in visual-based 3D object detection methods. We encourage future research in visual-related domains to further investigate our method and consider incorporating techniques such as frame-wise matching, as seen in methods like BEVStereo, to enhance depth prediction accuracy and further improve the overall accuracy of visual 3D object detection.

---

### Official Review · Reviewer_Qpuj · 2023-10-31

**Soundness:** 3 good
**Presentation:** 4 excellent
**Contribution:** 3 good
**Rating:** 6
**Confidence:** 3

**Summary:**

In this paper, the authors propose a novel sensor fusion technique that generates the fused BEV feature from LiDAR voxel features and image features without compressing the Z-axis information. Unlike prior works that generate separate point BEV features and image BEV features and then fuse them, the proposed method directly generates the fused BEV feature using queries in the BEV space and deformable attention modules to interact with point voxel features and 2D image features. For each BEV query, the authors generate multiple reference points with different heights and project them back to the voxel space or image feature space for deformable attention. Besides, a temporal fusion encoder is proposed to include temporal information. The proposed method can also be used for pure image BEV feature generation with additional depth estimation networks. Experiments show that the proposed method provides competitive performance with SOTA in 3D object detection.

**Strengths:**

1) The paper is well-written and well-organized.
2) The multi-modal fusion problem studied in this paper is interesting and timely in Autonomous Driving.
3) The proposed method is simple and interesting. It can be seen as an extension of BEVFormer in the multi-modal settings.

**Weaknesses:**

1) The proposed method is only tested in one dataset (nuScenes) and one task (3D object detection), which may not be enough to show the generalizability of the proposed sensor fusion scheme. It would be better to include more datasets (Waymo) and tasks (e.g., segmentation)

2) Though the proposed method makes use of the z-axis information, it seems to greatly increase the model complexity. To make fair comparisons with other baselines, the authors may better include complexity analysis such as FLOPS, #of parameters, and FPS

3) From Tables 1 and 2, the proposed method only provides a marginal improvement.

**Questions:**

In Table 1, the authors only show the result that combines temporal information. How about the one that only uses single-frame BEV features? (i.e., FusionFormer-S in Table 2). This result is important to evaluate the sensor fusion mechanism when compared with other CL baselines.

---

> ### Author Response · Authors · 2023-11-20
> **Thanks for your comments!**
>
> Thank you for your constructive comments and suggestions, and they are exceedingly helpful for us to improve our paper. We have carefully incorporated them in the revised paper. In the following, your comments are first stated and then followed by our point-by-point responses.
>
> ## Q1 More experiments.
> > The proposed method is only tested in one dataset (nuScenes) and one task (3D object detection), which may not be enough to show the generalizability of the proposed sensor fusion scheme. It would be better to include more datasets (Waymo) and tasks (e.g., segmentation)
>
> **Author response:**
> It is worth noting that previous studies on multimodal fusion have primarily focused on the nuScenes dataset, and there is little research conducted on the Waymo dataset. This may be attributed to the differences between the datasets since the Waymo dataset primarily consists of LiDAR data, and the available images predominantly capture a forward-facing view while lacking rearward images.
>
> We are trying to evaluate the generalizability of our method on the Waymo dataset and other tasks such as segmentation. The experiments are still ongoing. We will include relevant experimental results in the final version of our paper.
>
>
> ## Q2 Complexity comparison with other baselines
> > Though the proposed method makes use of the z-axis information, it seems to greatly increase the model complexity. To make fair comparisons with other baselines, the authors may better include complexity analysis such as FLOPS, #of parameters, and FPS
>
> **Author response:**
> Due to the constraints on the page limit of the main text, we present the complexity efficiency comparison in Appendix A.2.
>
> As shown in Table 7, we compare the efficiency of FusionFormer and existing methods. The efficiency and performance are tested on a single Tesla A100 GPU with the best model setting of official repositories. In comparison to BEVFusion, FusionFormer demonstrates superior performance with notable improvements of 3.1% in mAP and 2.7% in NDS, while maintaining a similar processing speed.
>
> *Appendix Table 7 Efficiency comparison on the nuScenes val set. ”L” is LiDAR. ”C” is camera. ”T” is temporal. The ”-S” indicates that the model only utilizes single-frame BEV features without incorporating temporal fusion techniques.*
> |Methods|Modality|mAP|NDS|FPS|
> |:----:|:----:|:----:|:----:|:----:|
> |TransFusion|CL|67.5|71.3|3.2|
> |BEVFusion|CL|68.5|71.4|4.2|
> |UVTR|CL|65.4|70.2|2.6|
> |CMT|CL|70.3|72.9|**6.0**|
> |DeepInteraction|CL|69.8|72.6|1.7|
> |FusionFormer-S|CL|70.0|73.2|4.0|
> |FusionFormer|CLT|**71.4**|**74.1**|3.8|
>
> In addition, we conducted a comparative experiment on computation cost between our method and the previous state-of-the-art method, CMT. As shown in Table 8, our method has similar FLOPS and parameters as CMT.
>
> *Appendix Table 8 Computation cost comparison on the nuScenes val set. ”L” is LiDAR. ”C” is camera. ”T” is temporal. The ”-S” indicates that the model only utilizes single-frame BEV features without incorporating temporal fusion techniques.*
> |Methods|Modality|mAP|NDS|FLOPS|Params|
> |:----:|:----:|:----:|:----:|:----:|:----:|
> |CMT|CL|70.3|72.9|2.17T|77.73M|
> |FusionFormer-S|CL|70.0|73.2|2.33T|77.55M|
> |FusionFormer|CLT|71.4|74.1|2.42T|78.54M|
>
>
> ## Q3 Marginal improvement
> > From Tables 1 and 2, the proposed method only provides a marginal improvement.
>
> **Author response:**
> The nuScenes 3D object detection leaderboard has currently reached a high level. In this scenario, our method still achieves overall performance improvement and significant improvement in certain aspects.
> - We achieves significant improvements in terms of reducing the **orientation error**（mAOE）and **classification error** (mAAE). Our method reduces the mAOE and mAAE compared to the previous state-of-the-art methods by 7.1% and 6.3%, respectively. These two errors are often considered challenging in point cloud-based detection. This indicates that our method is capable of better fusing image features to compensate for the shortcomings of point cloud features.
> - Our method exhibits better detection performance in scenarios where point clouds are sparse, particularly over long distances. This superior performance can be observed in Figure 6(b) of our paper.
>
> In summary, our method successfully achieves unified multimodal temporal fusion and delivers superior detection performance.

---

> > ### Author Response · Authors · 2023-11-20
> > **Q4 Single frame BEV feature result**
> >
> > ## Q4 Single frame BEV feature result
> > > In Table 1, the authors only show the result that combines temporal information. How about the one that only uses single-frame BEV features? (i.e., FusionFormer-S in Table 2). This result is important to evaluate the sensor fusion mechanism when compared with other CL baselines.
> >
> > **Author response:**
> > We present the results as requested on nuScenes test set. Note that we do not use any test-time augmentation. Our single-frame FusionFormer (FusionFormer-S) achieves competitive performance. Nonetheless, we emphasize we aim to propose a uniform architecture for multi-modal, temporal consistent architecture.
> >
> > *Performance comparison on the nuScenes test set. "L" is LiDAR. "C" is camera. "T" is temporal. The results are evaluated using a single model without any test-time-augmentation or ensembling techniques.*
> > |Methods|Modality|NDS↑|mAP↑|mATE↓|mASE↓|mAOE↓|mAVE↓|mAAE↓|
> > |:----:|:----:|:----:|:----:|:----:|:----:|:----:|:----:|:----:|
> > |PointPainting|CL|61.0|54.1|38.0|26.0|54.1|29.3|13.1|
> > |PointAugmenting|CL|71.1|66.8|25.3|23.5|35.4|26.6|12.3|
> > |MVP|CL|70.5|66.4|26.3|23.8|32.1|31.3|13.4|
> > |FusionPainting|CL|71.6|68.1|25.6|23.6|34.6|27.4|13.2|
> > |TransFusion|CL|71.7|68.9|25.9|24.3|35.9|28.8|12.7|
> > |BEVFusion|CL|72.9|70.2|26.1|23.9|32.9|26.0|13.4|
> > |BEVFusion|CL|73.3|71.3|**25.0**|24.0|35.9|25.4|13.2|
> > |UVTR|CL|71.1|67.1|30.6|24.5|35.1|**22.5**|12.4|
> > |CMT|CL|74.1|72.0|27.9|23.5|30.8|25.9|11.2|
> > |DeepInteraction|CL|73.4|70.8|25.7|24.0|32.5|24.5|12.8|
> > |BEVFusion4D-S|CL|73.7|71.9| - | - | - | - | - |
> > |BEVFusion4D|CLT|74.7|**73.3**| - | - | - | - | - |
> > |FusionFormer-S|CL|73.8|70.8|26.7|**23.4**|28.9|25.8|10.7|
> > |FusionFormer|CLT|**75.1**|72.6|26.7|23.6|**28.6**|**22.5**|**10.5**|

---

### Author Response · Authors · 2023-11-20
**To area chairs and all reviewers: Clarification of the novelty**

Thank you to all the reviewers for your time and constructive comments. In this response, we aim to address the concerns related to the innovation of our method. We would like to take this opportunity to emphasize the main contributions of our paper and highlight the distinguishing factors from existing related works, specifically BEVFormer and CMT.

## Our Contributions

**1. We advocate using point cloud features in __voxel format__ as opposed to using compressed LiDAR BEV features along the Z-axis.**

The recent approaches, BEVFusion, DeepInteraction and CMT, rely on LiDAR BEV features which is compressed along the Z-axis. Compared to fusing LiDAR features in the form of BEV features, our voxel feature fusion approach exhibits remarkable improvements in both object center position prediction and orientation prediction accuracy as shown in Table 5 in our paper (we provide the table here for your convenience).

*Study for the representation of the LiDAR feature on the nuScenes val-set, as in Table.*
|LiDAR|mAP↑|NDS↑|mATE↓|mAOE↓|
|:----:|:----:|:----:|:----:|:----:|
|BEV|61.3|66.1|35.7|36.9|
|Voxel|**62.7**|**67.3**|**34.4**|**31.4**|

**2. Based on our voxel feature fusion approach, we have discovered that it is possible to enhance visual-based 3D object detection by replacing LiDAR features with voxel features generated from *monocular depth predictions*.** This novel idea provides a new direction for improving the performance of visual-based 3D object detection.

*Results of camera based 3D detection on the nuScenes val set.*
|Method|Backbone|mAP|NDS|
|:----:|:----:|:----:|:----:|
|BEVFormer|R101-DCN|41.6|51.7|
|FusionFormer-Depth|R101-DCN|**43.9**|**53.5**|

**3. We have proposed a unified multimodal temporal fusion method based on Transformers, which has achieved a new state-of-the-art performance on the nuScenes dataset.**

## The distinguishing factors from existing related works.
### BEVFormer
- BEVFormer only supports **single-modality** visual input. In contrast, our method focuses on providing a novel **multimodal** fusion solution.
- BEVFormer adopts a recurrent-based temporal fusion approach that does not support long-term temporal fusion. In contrast, our method efficiently achieves **long-term temporal fusion**.


### DeepInteraction
- DeepInteraction focus on the **interaction** between multimodal features, while our method focuses on **modal fusion**.
- DeepInteraction preserves the original form of each modality feature after interaction, while our method generates **fused features**.
- DeepInteraction uses compressed **BEV lidar features**, while we preserve the Z-axis information by using **voxel features**.
- DeepInteraction relies on both modal inputs simultaneously, while our method exhibits **robustness** in the presence of **missing modalities**.
- DeepInteraction is not easily applicable for temporal fusion, while our method supports **temporal fusion**.


### CMT
- CMT **tokenizes** image and LiDAR features directly and uses a transformer decoder to generate object detection predictions, while our method generates **fused BEV features** that are then connected to the head for prediction.
- CMT uses compressed **BEV lidar features**, while we preserve the Z-axis information by using **voxel features**.
- CMT have challenges in achieving long term temporal fusion, while our method supports **long term temporal fusion**.

---

### Meta-Review · Area_Chair_od6A · 2023-12-13

**Metareview:**

This paper receives 3x marginally above the acceptance threshold and 1x reject, not good enough. Although the strength of the paper is that it proposes a method to solve the multi-modal fusion problem, which is interesting and timely for autonomous driving, there are several major concerns pointed out by the reviewers. The lack of novelty: The proposed framework has a large overlay with both DeepInteraction and Bevformer. Despite the clarification by the authors in the rebuttal, the reviewer remains unconvinced. It's also pointed out by another reviewer that the core idea of utilizing BEV queries for temporal and cross-modality fusion is widely used in previous methods like BEVFormer. The other reviewers pointed out that there's only one dataset used in the paper for comparison, which is insufficient for generalizability of the work. There is also concern on unfair comparison as most baselines only consider LiDAR and camera.

**Justification For Why Not Higher Score:**

The major concern is on the novelty of the proposed method and the inadequate experiment results.

**Justification For Why Not Lower Score:**

N/A

---

### Decision · Program_Chairs · 2024-01-16

Reject